# Latent Bandits Revisited

**Joey Hong**
Google Research
jxihong@google.com

**Branislav Kveton**
Google Research
bkveton@google.com

**Manzil Zaheer**
Google Research
manzilzaheer@google.com

**Yinlam Chow**
Google Research
yinlamchow@google.com

**Amr Ahmed**
Google Research
amra@google.com

**Craig Boutilier**
Google Research
cboutilier@google.com

## Abstract

A *latent bandit* is a bandit problem where the learning agent knows reward distributions of arms conditioned on an *unknown discrete latent state*. The goal of the agent is to identify the latent state, after which it can act optimally. This setting is a natural midpoint between online and offline learning, where complex models can be learned offline and the agent identifies the latent state online. This is of high practical relevance, for instance in recommender systems. In this work, we propose general algorithms for latent bandits, based on both upper confidence bounds and Thompson sampling. The algorithms are contextual, and aware of model uncertainty and misspecification. We provide a unified theoretical analysis of our algorithms, which have lower regret than classic bandit policies when the number of latent states is smaller than actions. A comprehensive empirical study showcases the advantages of our approach.

## 1 Introduction

Many online platforms, such as search engines and recommender systems, display results based on observed properties of the user and their query. However, the user's behavior is often influenced by a *latent state* not explicitly revealed to the system. This might be *user intent* (e.g., reflecting a long-term task) in search, or *short- and long-term preferences* (e.g., reflecting topic interests) in a recommender. In each case, the unobserved latent state influences the user response to the displayed results, and thus the associated reward. A machine learning (ML) system, thus, should take steps to infer the latent state and tailor its results accordingly.

While many ML models use either heuristic features [1, 4] or recurrent models [33] to capture user history, explicit exploration for *latent state identification* (i.e., reducing uncertainty regarding the true state) is less common in practice. In this paper, we study *latent bandits*, which model online interactions of the above type as follows. In each round, the learning agent observes context (e.g., query or user demographics), takes an action (e.g., a recommends), and observes its reward (e.g., user engagement with the recommendation). The reward depends stochastically on both the context and user latent state. As a result, it provides information about the unobserved latent state, which can be used to improve predictions in the future. We are interested in designing exploration policies that allow the agent to quickly maximize its per-round reward by resolving the *relevant* latent state uncertainty. Specifically, we want policies that have low $n$-*round regret*.

Our latent state structure allows an agent to quickly adapt its results to new users, or adapt to new user tasks or intents on a per-session basis. One example of that structure are clusters of users with similar item preferences [35]. In this example, a system should be able to identify which cluster a

new user belongs to faster than learning all user preferences without any prior information. This would yield much better recommendations in the short-horizon, or cold-start, regime.

*Fully online exploration* (e.g., for personalization) involves learning a reward model, conditioned on the context and latent state, and generally requires massive amounts of interaction data. Fortunately, many platforms have such *offline* data (e.g., past user interactions) available, which can be used to construct both a reasonable latent state space and conditional reward models [23, 9]. We assume that such models are available and focus on a simpler online problem of latent state identification. Prior works in this setting [25, 35] assumed that the *true* conditional reward models are given. Moreover, they focused on upper confidence bound (UCB) designs, which are theoretically optimal but often perform poorly empirically. We provide a unified framework that combines offline-learned models with online exploration using both UCBs and Thompson sampling. Our algorithms are practical, analyzable, contextual, and robust to natural forms of model misspecification.

We are the first to propose algorithms for latent bandits that are aware of model uncertainty. Our paper is organized as follows. In Section 3, we propose novel practical algorithms based on UCBs and Thompson sampling. Based on a connection between UCBs and posterior sampling [28], we derive near-optimal bounds on the Bayes regret of our algorithms in Section 4. Finally, in Section 5, we demonstrate their effectiveness in synthetic simulations and on a large-scale real-world dataset.

## 2  Problem Formulation

We adopt the following notation. Random variables are capitalized. The set of actions is $\mathcal{A}$, the set of contexts is $\mathcal{X}$, and the set of latent states is $\mathcal{S}$, with $|\mathcal{S}| \ll |\mathcal{A}|$.

We study a *latent bandit* problem, where the learning agent interacts with an environment over $n$ rounds as follows. In round $t \in [n]$, the agent observes context $X_t \in \mathcal{X}$, takes action $A_t \in \mathcal{A}$, and observes reward $R_t \in \mathbb{R}$. The random variable $R_t$ depends on the context $X_t$, action $A_t$, and latent state $s \in \mathcal{S}$, where $s$ is fixed but unknown.[1] The *observation history* of the learning agent up to round $t$ is a vector $H_t = (X_1, A_1, R_1, \ldots, X_{t-1}, A_{t-1}, R_{t-1})$. The *policy* of the agent maps $H_t$ and $X_t$ to the choice of action $A_t$.

Now we state our assumptions on how rewards and contexts are generated. The *rewards* are drawn i.i.d. from a *conditional reward distribution* $P(\cdot \mid A, X, s; \theta)$, which is parameterized by a vector $\theta \in \Theta$, where $\Theta$ is a set of feasible model parameters. We use $\mu(a, x, s; \theta) = \mathbb{E}_{R \sim P(\cdot|a,x,s;\theta)}[R]$ to denote the *mean reward* of action $a$ in context $x$ and latent state $s$ under model $\theta$. We denote the true (unknown) latent state by $s_*$ and true model parameters by $\theta_*$. We assume that $\theta_*$ can be *estimated offline*. We also assume that the rewards are $\sigma^2$-sub-Gaussian, that is

$$\mathbb{E}_{R \sim P(\cdot|a,x,s;\theta_*)}\left[\exp(\lambda(R - \mu(a, x, s; \theta_*)))\right] \leq \exp(\sigma^2 \lambda^2 / 2)$$

for all $a$, $x$, $s$, and $\lambda > 0$. The contexts can be generated by an arbitrary process independent of the agent's actions and the mean reward $\mu(a, x, s; \theta)$ can be any complex function of $\theta$.

The performance of the agent is measured by regret. Let $A_{t,*} = \arg\max_{a \in \mathcal{A}} \mu(a, X_t, s_*; \theta_*)$ be the optimal action for latent state $s_* \in \mathcal{S}$ and model $\theta_* \in \Theta$. Then the *expected $n$-round regret* is

$$\mathcal{R}(n; s_*, \theta_*) = \mathbb{E}\left[\sum_{t=1}^{n} \mu(A_{t,*}, X_t, s_*; \theta_*) - \mu(A_t, X_t, s_*; \theta_*)\right]. \tag{1}$$

While a fixed-state regret is useful, we are often more concerned with the average performance over a range of states (e.g., multiple users and multiple sessions with the same user). Thus we also consider the Bayes regret, where we take an expectation over a random latent state and model. Suppose that $S_*$ and $\theta_*$ are drawn from some prior $P_1$. Then the *$n$-round Bayes regret* is

$$\mathcal{BR}(n) = \mathbb{E}\left[\mathcal{R}(n; S_*, \theta_*)\right] = \mathbb{E}\left[\sum_{t=1}^{n} \mu(A_{t,*}, X_t, S_*; \theta_*) - \mu(A_t, X_t, S_*; \theta_*)\right]. \tag{2}$$

Note that $S_*$ and $\theta_*$ in the definition of $A_{t,*} = \arg\max_{a \in \mathcal{A}} \mu(a, X_t, S_*; \theta_*)$ are random now.

**Algorithm 1** mUCB

---

1: **Input:** Model parameters $\widehat{\theta}$

2: **for** $t \leftarrow 1, 2, \ldots$ **do**
3:    Define $N_t(s) \leftarrow \sum_{\ell=1}^{t-1} \mathbb{1}\{B_\ell = s\}$ and

$$G_t(s) \leftarrow \sum_{\ell=1}^{t-1} \mathbb{1}\{B_\ell = s\} \left(\widehat{\mu}(A_\ell, X_\ell, s) - R_\ell\right) \tag{3}$$

4:    Set of consistent latent states $C_t \leftarrow \left\{s \in \mathcal{S} : G_t(s) \leq \sigma \sqrt{6 N_t(s) \log n}\right\}$
5:    Select $B_t, A_t \leftarrow \arg\max_{s \in C_t, a \in \mathcal{A}} \widehat{\mu}(a, X_t, s)$

---

## 3 Algorithms

In this section, we develop both UCB and Thompson sampling (TS) algorithms that leverage an arbitrarily complex environment model, generally learned offline, to expedite online exploration. As discussed earlier, such offline models can be readily learned given large amounts of offline interaction data available in many interactive systems. In each subsection below, we specify a particular form of the offline-learned model, and develop a corresponding online algorithm. It is important to note that given an environment model, an optimal solution is to compute and maximize the Gittins index over actions [15]. However, this is often computationally intractable and does not generalize to complex latent variable models. We want our methods to be practical, and thus only consider myopic policies that can be efficiently implemented online.

### 3.1 UCB with Perfect Model (mUCB)

We first design a UCB-like algorithm with learned model $\widehat{\theta} \in \Theta$. Let $\widehat{\mu}(a, x, s) = \mu(a, x, s; \widehat{\theta})$ be the mean reward under model $\widehat{\theta}$ and $\mu(a, x, s) = \mu(a, x, s; \theta_*)$ be the true reward. We initially assume that the learned model is accurate, that is we are given $\widehat{\theta} = \theta_*$ as an input.

The key idea in UCB algorithms is to compute a high-probability upper confidence bound $U_t(a)$ on the mean reward of each action $a$ in any round $t$, where the $U_t$ is some function of history [8]. Then the algorithms take action $A_t = \arg\max_{a \in \mathcal{A}} U_t(a)$. Our model-based UCB algorithm, which we call mUCB, works very similarly. The pseudocode of mUCB is in Algorithm 1. The algorithm works as follows. In round $t$, it maintains a set of latent states $C_t$ that are *consistent* with the observed rewards thus far. Then it chooses a specific *believed latent state* $B_t$ from the consistent set $C_t$ and takes action $A_t$ with the maximum expected reward in that state, $(B_t, A_t) = \arg\max_{s \in C_t, a \in A} \widehat{\mu}(a, X_t, s)$. Thus the UCB of action $a$ is $U_t(a) = \arg\max_{s \in C_t} \widehat{\mu}(a, X_t, s)$. mUCB tracks two key quantities: the number of times that state $s$ is selected up to round $t$, $N_t(s)$; and the "gap" between the expected and realized rewards in state $s$ up to round $t$, $G_t(s)$. If $G_t(s)$ is high, mUCB marks state $s$ as *inconsistent* and does not consider it in round $t$. Note that the gap is defined over latent states rather than actions, and with respect to realized rewards rather than expected rewards.

### 3.2 UCB with Misspecified Model (mmUCB)

Now we extend mUCB to misspecified models, where we are given $\widehat{\theta} \neq \theta_*$ as an input. We consider the following worst-case high-probability definition of *model misspecification*: there exist $\varepsilon, \delta > 0$ such that $|\widehat{\mu}(a, x, s) - \mu(a, x, s)| \leq \varepsilon$ holds with probability at least $1 - \delta$ jointly for all $a$, $x$, and $s$. Guarantees of this form, for example, exist for spectral methods for latent variable models, where $\varepsilon$ and $\delta$ depend on the size of the offline dataset [5].

We modify mUCB to be robust to this type of model error, which gives rise to a new algorithm mmUCB. The only change to mUCB is that $G_t(s)$ is replaced with a high-probability lower bound

$$G_t(s) = \sum_{\ell=1}^{t-1} \mathbb{1}\{B_\ell = s\} \left(\widehat{\mu}(A_\ell, X_\ell, s) - \varepsilon - R_\ell\right) . \tag{4}$$

| **Algorithm 2** mTS | **Algorithm 3** mmTS |
|---|---|
| 1: **Input:** | 1: **Input:** |
| 2:     Model parameters $\widehat{\theta}$ | 2:     Prior over latent states and model parameters $P_1(s,\theta)$ |
| 3:     Prior over latent states $P_1(s)$ | |
| 4: **for** $t \leftarrow 1, 2, \dots$ **do** | 3: **for** $t \leftarrow 1, 2, \dots$ **do** |
| 5:     Define | 4:     Define |
|     $P_t(s) \propto P_1(s) \prod_{\ell=1}^{t-1} P(R_\ell \mid A_\ell, X_\ell, s; \widehat{\theta})$ |     $P_t(s,\theta) \propto P_1(s,\theta) \prod_{\ell=1}^{t-1} P(R_\ell \mid A_\ell, X_\ell, s; \theta)$ |
| 6:     Sample $B_t \sim P_t$ | 5:     Sample $B_t, \widehat{\theta} \sim P_t$ |
| 7:     Select $A_t \leftarrow \arg\max_{a \in \mathcal{A}} \widehat{\mu}(a, X_t, B_t)$ | 6:     Select $A_t \leftarrow \arg\max_{a \in \mathcal{A}} \widehat{\mu}(a, X_t, B_t)$ |

This allows mmUCB to be conservative when identifying inconsistent latent states, so that $s_* \in C_t$ remains to hold with a high probability. Just as importantly, it is also useful for deriving worst-case regret bounds, both for UCB algorithms and TS.

### 3.3 Thompson Sampling with Perfect Model (mTS)

Our UCB algorithms are designed for the worst case. Now we adopt a more relaxed design, where the latent state is random, and the model is fixed and known. That is, we are given $\widehat{\theta} = \theta_*$ and a prior distribution over latent states $P_1$ as inputs.

Our solution is a variant of Thompson sampling [32, 10, 28]. The key idea in TS is to sample actions according to their posterior probability of being optimal, conditioned on the history of the agent. In our case, the optimal action in round $t$ is $A_{t,*} = \arg\max_{a \in A} \mu(a, X_t, S_*; \theta_*)$, and is random both due to the observed context $X_t$ and unknown latent state $S_*$. Therefore, TS should take action $a$ with probability $\mathbb{P}(A_t = a \mid X_t, H_t) = \mathbb{P}(A_{t,*} = a \mid X_t, H_t)$. An advantage of TS over UCB algorithms is that it obviates the need to design UCBs, which are often loose. As a result, UCB algorithms are often conservative in practice and TS typically offers better empirical performance [10].

Our model-based TS algorithm, which we call mTS, is presented in Algorithm 2. The algorithm works as follows. In round $t$, it maintains a posterior probability, $P_t(s) = \mathbb{P}(S_* = s \mid H_t)$, that each latent state $s$ is optimal. Then it samples the latent state from its posterior distribution, $B_t \sim P_t$, and takes action $A_t = \max_{a \in \mathcal{A}} \widehat{\mu}(a, X_t, B_t)$. For any fixed $s$, $P_t(s) \propto P_1(s) \prod_{\ell=1}^{t-1} P(R_\ell \mid A_\ell, X_\ell, s; \widehat{\theta})$. As a result, $P_t$ can be updated incrementally in the standard Bayesian filtering fashion [30]. Note that unlike mUCB, mTS needs to know the conditional reward distribution $P(\cdot \mid a, x, s; \widehat{\theta})$ to update $P_t(s)$.

### 3.4 Thompson Sampling with Misspecified Model (mmTS)

Now we extend mTS to misspecified models. However, instead of considering a worst-case estimate of $\theta_*$, as in mmUCB, we assume that $\theta_* \sim P_1$ and that the learning agent knows $P_1$. This is motivated by prior literature on modeling epistemic uncertainty [11]. In practice, learning a distribution over parameters is intractable for complex models; but approximate inference can be always performed, for instance by an ensemble of bootstrapped models [11].

Our TS algorithm for misspecified models, which we call mmTS, is presented in Algorithm 3. The algorithm seamlessly integrates model uncertainty into mTS as follows. In round $t$, the latent state $B_t$ and estimated model parameters $\widehat{\theta}$ are sampled from their joint posterior, and then an action is taken to maximize $A_t = \max_{a \in \mathcal{A}} \widehat{\mu}(a, X_t, B_t)$. In general, sequential Monte Carlo methods [12] can be used for approximate posterior sampling. However, when the model prior is conjugate to the likelihood, the posterior has a closed form and can be sampled from as follows. Since $\mathcal{S}$ is finite, we can tractably sample from the joint posterior by first sampling latent state $B_t$ from its marginal posterior and then $\widehat{\theta}$ conditioned on $B_t$. For exponential family distributions, the posterior parameters can also be updated online and efficiently. We provide more details in Appendix A, and a pseudocode for Gaussians in Appendix B.

# 4 Regret Analysis

Maillard and Mannor [25] derived gap-dependent regret bounds for UCB algorithms in latent bandits under the assumption that the true model is known and arms are independent. We provide a unified analysis that extends their results to include context, model misspecification, and TS algorithms.

## 4.1 Regret Decomposition

UCB algorithms explore using upper confidence bounds, while TS samples from the posterior. Russo and Van Roy [28] related these two designs with a unified regret decomposition. In our problems, this is reflected as follows. Let $s_*$ be the true latent state. Then the regret of our UCB algorithms in round $t$ decomposes as

$$\mu(A_{t,*}, X_t, s_*) - \mu(A_t, X_t, s_*) = \mu(A_{t,*}, X_t, s_*) - U_t(A_t) + U_t(A_t) - \mu(A_t, X_t, s_*)$$
$$\leq [\mu(A_{t,*}, X_t, s_*) - U_t(A_{t,*})] + [U_t(A_t) - \mu(A_t, X_t, s_*)] ,$$

where the inequality holds by the definition of $A_t$. A similar inequality without latent states appears in prior work [28]. This yields the following regret decomposition

$$\mathcal{R}(n; s_*, \theta_*) \leq \mathbb{E}\left[\sum_{t=1}^{n} \mu(A_{t,*}, X_t, s_*) - U_t(A_{t,*})\right] + \mathbb{E}\left[\sum_{t=1}^{n} U_t(A_t) - \mu(A_t, X_t, s_*)\right] . \quad (5)$$

An analogous decomposition exists for the Bayes regret of our TS algorithms. Specifically, for any TS algorithm and function $U_t$ of history, we have

$$\mathcal{BR}(n) = \mathbb{E}\left[\sum_{t=1}^{n} \mu(A_{t,*}, X_t, S_*; \theta_*) - U_t(A_{t,*})\right] + \mathbb{E}\left[\sum_{t=1}^{n} U_t(A_t) - \mu(A_t, X_t, S_*; \theta_*)\right] . \quad (6)$$

The proof uses the fact that $\mathbb{E}[U_t(A_{t,*}) \mid X_t, H_t] = \mathbb{E}[U_t(A_t) \mid X_t, H_t]$ holds for any $H_t$ and $X_t$ from the design of TS. Thus $U_t$ can be an upper confidence bound in a UCB algorithm.

Though the UCBs $U_t$ are not used by TS, they can be used to *analyze* it, due to the similarity of (5) and (6). Thus regret bounds for UCB algorithms can be translated to Bayes regret bounds for TS. There are two caveats though. First, since actions in TS do not maximize $U_t$, the regret bound must be proved using a worst-case argument over suboptimal actions. Second, because the Bayes regret is in expectation over problem instances, the resulting regret bounds are problem-independent, also known as gap-free.

## 4.2 Key Steps in Our Proofs

Full proofs of our regret bounds are in Appendix C. All proofs follow the same outline, the key steps of which are outlined below. To ease the exposition, we assume that the suboptimality gap of any action is bounded by 1.

**Step 1: Concentration of realized rewards at their means.** We first show that the total observed reward does not deviate too much from its expectation, under any believed latent state $s$. Formally, we show that $\mathbb{P}\left(\left|\sum_{\ell=1}^{t-1} \mathbb{1}\{B_\ell = s\} (\mu(A_\ell, X_\ell, s_*) - R_\ell)\right| \geq \sigma\sqrt{6N_t(s)\log n}\right) = O(n^{-2})$ holds for any round $t$ and state $s$. Since we consider the contextual case, which requires joint estimation over dependent arms, we use martingales and Azuma's inequality.

**Step 2: $s_* \in C_t$ in each round $t$ with a high probability.** This follows from the definition of $C_t$ and the concentration argument in Step 1 for $s = s_*$. Then, in any round $t$ where $s_* \in C_t$, we can use that $U_t(a) \geq \mu(a, X_t, s_*)$ for any action $a$ in mUCB, and $U_t(a) \geq \mu(a, X_t, s_*) - \varepsilon$ in mmUCB.

**Step 3: Upper bound on the UCB regret.** We prove this by bounding each term in (5) separately. The first term is at most 0 with a high probability by Step 2. The second term is a sum of confidence widths, the differences between $U_t$ and the mean reward in round $t$. We partition it by the chosen latent state in each round. For each latent state $s$, we introduce realized rewards $R_t$ and get

$$\sum_{t=1}^{n} \mathbb{1}\{B_t = s\} (U_t(A_t) - \mu(A_t, X_t, s_*)) \leq G_n(s) + 1 + \sum_{t=1}^{n} \mathbb{1}\{B_t = s\} (R_t - \mu(A_t, X_t, s_*)).$$

The key step in proving the above bound is that $G_\ell(s) = G_n(s)$ holds in the last round $\ell$ where state $s$ is chosen, where $\sum_{t=1}^{\ell-1} \mathbb{1}\{B_t = s\}(U_t(A_t) - R_t) \leq G_\ell(s)$ holds. This relation also implies that $G_n(s) \leq \sigma\sqrt{6N_n(s)\log n}$. The other term is bounded by Step 1, which gives a total upper bound of $2\sigma\sqrt{6N_n(s)\log n}$. Finally, we apply the Cauchy-Schwarz inequality to combine the bounds for individual latent states.

**Step 4: Upper bound on the TS regret.** We use the fact that the regret decomposition for Bayes regret in (6) is similar to that for the UCB regret in (5). As mentioned in Section 4.1, as long as our UCB analysis in Step 3 is worst-case over all possible sequences of actions and gap-free, the UCB regret bound transfers to a Bayes regret bound for TS.

### 4.3 Regret Bounds

Our first result is an upper bound on the $n$-round regret of mUCB. This result differs from Maillard and Mannor [25] in two respects: our bound is gap-free and accounts for context.

**Theorem 1.** *Assume that $\widehat{\theta} = \theta_*$. Then, for any $s_* \in \mathcal{S}$ and $\theta_* \in \Theta$, the $n$-round regret of mUCB is bounded as $\mathcal{R}(n; s_*, \theta_*) \leq 2\sigma\sqrt{6|\mathcal{S}|n\log n} + 3|\mathcal{S}|$.*

A gap-free lower bound on the regret in a $K$-armed bandit is $\Omega(\sqrt{Kn})$ [7]. So our upper bound is optimal up to log factors, when substituting actions $\mathcal{A}$ with latent states $\mathcal{S}$. The bound can be much lower when $|\mathcal{S}| \ll K$. It also holds for arbitrary reward models and is contextual. From Step 4 of the proof outline, we also have that the Bayes regret of mTS is bounded.

**Corollary 1.** *Assume that $\widehat{\theta} = \theta_*$. Then, for $S_* \sim P_1$ and any $\theta_* \in \Theta$, the $n$-round Bayes regret of mTS is bounded as $\mathcal{BR}(n) \leq 2\sigma\sqrt{6|\mathcal{S}|n\log n} + 3|\mathcal{S}|$.*

Our next results apply to misspecified models. We assume that $\widehat{\theta}$ is estimated by some black-box method. For mmUCB, our bound depends on a high-probability maximum error $\varepsilon$ induced by $\widehat{\theta}$.

**Theorem 2.** *Let $\mathbb{P}\left(\forall a \in \mathcal{A}, x \in \mathcal{X}, s \in \mathcal{S} : |\mu(a, x, s; \widehat{\theta}) - \mu(a, x, s; \theta_*)| \leq \varepsilon\right) \geq 1 - \delta$ for some $\varepsilon, \delta > 0$. Then, for any $s_* \in \mathcal{S}$ and $\theta_* \in \Theta$, the $n$-round regret of mmUCB is bounded as*

$$\mathcal{R}(n; s_*, \theta_*) \leq 2\sigma\sqrt{6|\mathcal{S}|n\log n} + 2\varepsilon n + \delta n + 3|\mathcal{S}|.$$

The proof of Theorem 2 follows our outline in Section 4.2. Steps 1–2 remain unchanged, but Step 3 changes to account for the error due to model misspecification. The linear dependence on error $\varepsilon$ and probability $\delta$ is unavoidable in the worst case, specifically when $\varepsilon$ is larger than the suboptimality gap. Nevertheless, some offline learning methods, such as spectral methods for latent variable models [5], allow $\varepsilon$ and $\delta$ to be arbitrarily small as the size of the offline dataset grows. So our bound can be sublinear in $n$.

For mmTS, we assume that a prior distribution over model parameters is known. The key step in the proof is to introduce $\bar{\mu}(a, x, s) = \int_\theta \mu(a, x, s, \theta)P_1(\theta)d\theta$, the conditional mean reward marginalized over the prior. Using this quantity, we can obtain the following Bayes regret bound.

**Corollary 2.** *Let $\theta_* \sim P_1$ and $\mathbb{P}(\forall a \in \mathcal{A}, x \in \mathcal{X}, s \in \mathcal{S} : |\bar{\mu}(a, x, s) - \mu(a, x, s; \theta_*)| \leq \varepsilon) \geq 1 - \delta$ for some $\varepsilon, \delta > 0$. Then, for $S_*, \theta_* \sim P_1$, the $n$-round Bayes regret of mmTS is bounded as*

$$\mathcal{BR}(n) \leq 2\sigma\sqrt{6|\mathcal{S}|n\log n} + 2\varepsilon n + \delta n + 3|\mathcal{S}|.$$

The proof of Corollary 2 relies on a variant of mmUCB where $\widehat{\mu}(a, x, s)$ is replaced with $\bar{\mu}(a, x, s)$. Note that unlike in mmUCB, the linear dependence of $\varepsilon$ is conservative, as mmTS updates its model posteriors and eventually concentrates. We leave the derivation of a tighter bound for future work.

We can relate the error $\varepsilon$ and its probability $\delta$ using the tails of the conditional reward distributions. In particular, let $\mu(a, x, s; \theta) - \bar{\mu}(a, x, s)$ be $v^2$-sub-Gaussian in $\theta \sim P_1$ for any $a$, $x$, and $s$. Then for any $\delta \in [0, 1]$, $\varepsilon = O(v\sqrt{2\log(K|\mathcal{X}||\mathcal{S}|/\delta)})$ satisfies the conditions on $\varepsilon$ and $\delta$ in Corollary 2.

## 5 Experiments

In this section, we evaluate our algorithms on both synthetic and real-world datasets. We compare the following methods: (i) **UCB**: UCB1/LinUCB with no offline model [8, 1]; (ii) **TS**: TS/LinTS with

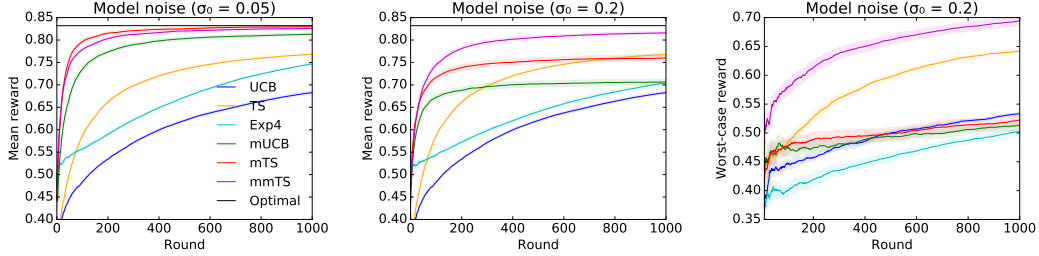

Figure 1: Left: Mean reward and standard error for low model noise ($\sigma_0 = 0.05$). Middle/Right: Mean/worst-case reward and standard error for high model noise ($\sigma_0 = 0.2$).

no offline model [3, 4]; (iii) **Exp4**: Exp4 where models are experts [7] (iv) **mUCB, mmUCB**: our proposed UCB algorithms mUCB/mmUCB; (v) **mTS, mmTS**: our proposed TS algorithms mTS/mmTS. In contrast to our methods, the UCB and TS baselines do not use an offline learned model. Exp4 uses models as experts, where each expert pulls the best arm given context and its latent state. Since we are interested in "fast personalization", we experiment with short horizons of up to 1000 rounds.

## 5.1 Synthetic Experiments

We first experiment with synthetic non-contextual bandits where $\mathcal{A} = [10]$ and $\mathcal{S} = [5]$. The mean rewards are sampled uniformly at random as $\mu(a, s) \sim \text{Uniform}(0, 1)$ for each $a \in \mathcal{A}, s \in \mathcal{S}$. Using rejection sampling, we constrain the suboptimality gap of all actions in each state $s$ to be at least 0.1, to ensure statistically significant comparisons at short horizons. The rewards are drawn i.i.d. from $P(\cdot \mid a, s) = \mathcal{N}(\mu(a, s), \sigma^2)$ with $\sigma = 0.5$. We evaluate each algorithm on 500 independent runs, with a uniformly sampled latent state in each run, and report the average reward over time. We analyze the effect of model misspecification by perturbing the mean rewards with various degrees of noise: for noise $\sigma_0 > 0$, the estimated means are sampled as $\widehat{\mu}(a, s) \sim \mathcal{N}(\mu(a, s), \sigma_0^2)$ for each action $a$ and latent state $s$.

The left plot in Figure 1 shows the average reward over time for low model noise, $\sigma_0 = 0.05$. In this setting, our algorithms mUCB and mTS perform better than the baselines UCB1 and TS. In the middle plot, we increase the noise to $\sigma_0 = 0.2$. Neither mUCB nor mTS accounts for modeling errors, and thus their performance degrades. On the other hand, the uncertainty-aware mmTS outperforms mTS. However, mmUCB (not reported to reduce clutter) performs the same as mUCB. This is likely because of the conservative nature of UCBs. Despite having similar worst-case regret guarantees, Exp4 is tailored to adversarial bandits instead of the stochastic ones. Therefore, it is outperformed by mUCB and mTS with the same model.

The right plot in Figure 1 shows $10\%$ of the worst runs from the middle plot, as measured by the average reward in the last round. This is a measure of a "worst-case" performance. Both UCB1 and TS are unaffected by model misspecification, and outperform mUCB and mTS. However, mmTS still performs best due to adapting the misspecified model to online data. This shows that employing uncertainty-awareness makes model-based algorithms much more robust to model misspecification and learning error.

## 5.2 MovieLens Experiments

We also assess the performance of our algorithms on the MovieLens 1M dataset [17], a large-scale collaborative filtering dataset where 6040 users rate 3883 movies. Each movie has a set of genres. We filter the dataset to include only users who rated at least 200 movies, and movies rated by at least 200 users; resulting in 1353 users and 1124 movies.

We randomly select $50\%$ of all ratings as our training set and use the remaining $50\%$ as the test set; resulting in sparse rating matrices $M_{\text{train}}$ and $M_{\text{test}}$. We complete each matrix using least-squares matrix completion [29] with rank 20. This rank is high enough to yield a low prediction error, and yet small enough to avoid overfitting. The learned latent factors are $M_{\text{train}} = \widehat{U}\widehat{V}^T$ and $M_{\text{test}} = UV^T$. User $i$ and movie $j$ correspond to rows $U_i$ and $V_j$, respectively, in the matrix factors.

We define a latent contextual bandit instance with $\mathcal{A} = [20]$ and $\mathcal{S} = [5]$ as follows. Using $k$-means clustering on the rows of $U$, we cluster users into 5 clusters, where 5 is the largest value that does not yield empty clusters. First, a user $i$ is sampled uniformly at random. In each round, 20 genres

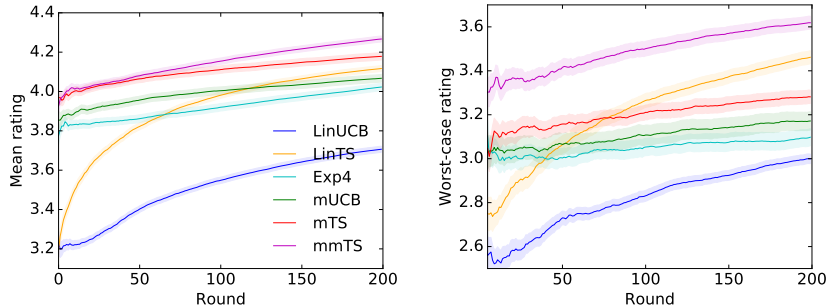

Figure 2: Mean/worst-case rating and standard error on the MovieLens 1M dataset.

and then a movie for each genre are uniformly sampled, creating a set of diverse movies. Context $X_t \in \mathbb{R}^{20 \times 20}$ is a matrix where each row is a training latent factor of one sampled movie, that is $\widehat{V}_j$ for movie $j$. The agent selects one movies from $X_t$ and observes its reward. The reward distribution for user $i$ and movie $j$ is $\mathcal{N}(U_i^T V_j, 0.5)$, and its mean is the product of the corresponding user and item factors on the test set. We evaluate on $500$ users.

Our "offline" model is a Gaussian mixture that is learned in the same way as the true model, except on the training set. It has $5$ clusters of users derived from $k$-means clustering on the rows of $\widehat{U}$. For each latent state, the prior is a Gaussian with the corresponding cluster mean and covariance. The context in LinUCB and LinTS is $X_t$, the same as in our algorithms, and they only learn the user latent factor. This is more information than in low-rank bandit algorithms [24], which learn both the user and movie representations, and thus perform poorly in short horizons that we consider.

The left plot in Figure 2 shows mean ratings and standard errors of $6$ algorithms (as earlier, mmUCB performs similarly to mUCB and is not shown). mUCB and mTS clearly adapt and "personalize" to users faster than LinUCB and LinTS, and converge to better policies than Exp4. Both mUCB and mTS are affected by model misspecification. In comparison, mmTS handles model uncertainty and converges to the best policies. The right plot in Figure 2 shows the average results for the worst $10\%$ of users. Again, mmTS dramatically outperforms mTS in the worst case.

## 6 Related Work

**Latent bandits.** Latent contextual bandits can personalize faster than standard contextual bandit policies, such as LinUCB [1] or linear Thompson sampling [4, 2]. The closest to our work is that of Maillard and Mannor [25], who proposed and analyzed non-contextual UCB algorithms, with side information that uniquely identifies users, under the assumption that the mean rewards for each latent state are known. Then they relaxed the assumption on known means, but assumed the other extreme case where the means are learned completely online. Zhou and Brunskill [35] extended this formulation to contextual bandits. However, they used offline-learned policies that were deployed online as a mixture, using Exp4. Bayesian policy reuse (BPR) [27] selects offline-learned policies by maintaining a belief over the optimality of each policy through posterior inference, but no analysis exists. Krause and Guestrin [21] used inference in latent graphical models to gather information, but only to identify the state. This is akin to best-arm identification [6], which is a different objective from cumulative regret minimization. We subsume prior work by providing contextual uncertainty-aware UCB and TS algorithms, and their unified analyses.

**Low-rank bandits.** Low-rank bandits can be viewed as a generalization of latent bandits, where low-rank matrices parameterize the reward and are learned online. Kawale *et al.* [20] proposed a TS algorithm for low-rank matrix factorization; however, their algorithm is inefficient and is analyzed only for rank-$1$ matrices. Sen *et al.* [31] analyzed an $\varepsilon$-greedy algorithm, but relied on the properties that rarely hold in practice. Another body of work studied online clustering of bandits, which is based on a specific low-rank structure [24, 13, 14, 26]. Yet another studied low-rank matrices where both rows and columns are arms [19, 18]. None of these prior works used offline-learned models, an important practical consideration given the general availability of offline data, and learned at most linear models. In Section 5, we compare to idealized versions of these methods where low-rank features are provided.

**Structured bandits.** In structured bandits, the arms are related through a common latent parameter. Lattimore and Munos [22] proposed a UCB algorithm for a multi-armed bandit variant of this problem. Recently, Gupta *et al.* [16] proposed a unified framework that adapts classic bandit algorithms, such as `UCB1` and TS, to structured multi-armed bandits. Though similar to our work, the algorithms differ in key aspects: we put confidence intervals on latent states instead of arms, and develop contextual algorithms with a prior model. Yu *et al.* [34] proposed variational Thompson sampling for learning to act in graphical models with latent variables. Since this framework is very general, both model parameters and latent variables are learned, the regret guarantees are weak and Thompson sampling needs to be approximated. This is in a stark contrast to our work, where we obtain strong regret guarantees and can implement algorithms as analyzed.

## 7    Conclusions

We study latent bandits, a class of bandit problems where the reward model is parameterized by a latent state and at least partially known. We propose UCB and Thompson sampling algorithms for solving this problem, which identify the latent state conditioned on offline-learned reward models. The algorithms are contextual and robust to misspecification. We bound their regret using a unified analysis, and validate them empirically on both a synthetic problem and the MovieLens 1M dataset.

Because of its generality and practicality, our work can be naturally extended to more complicated graphical models of the environment. For example, in this work, we assume that the latent state is fixed over time. However, a transition model could be incorporated into our setting to model an evolving latent state. This would be useful when user preferences or intents change over time. In addition, we could consider a more expressive latent state, such as a mixture of topics, which would be useful if user types could not be classified into a finite discrete set. We leave the detailed study of these extensions to future work.

## Broader Impact

Our work develops improved algorithms for bandit-style exploration in a very general and abstract sense. We have demonstrated its ability to increase the rate at which interactive systems identify user latent state to improve long-term impact on user reward (e.g., engagement in a recommender system). Our work is agnostic to the form of the reward. We are strongly motivated by improving user positive engagement with interactive systems (e.g., by identifying user interests or preferences in a recommender system). However, other forms of reward that are unaligned with a user's best interests could be used—our methods do not propose specific reward models. That said, our work has no social implications (welfare, fairness, privacy, etc.) beyond those already at play in the interactive system to which our methods might be applied.

## Acknowledgments and Disclosure of Funding

There are no additional sources of funding related to this work.

## Footnotes

[1]The latent state $s$ can be viewed as a user's current task or preferences, and is fixed over all $n$ rounds.

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
