[Supplementary Material]

# A Details of `mmTS` for Exponential Families

For a matrix (vector) $M$, we let $M_i$ denote its $i$-th row (element). Using this notation, we can write $\theta = (\theta_s)_{s \in \mathcal{S}}$ as a vector of parameters, one for each latent state; each $\theta_s$ parameterizes the reward under latent state $s$. We want to show that the sampling step in `mmTS` can be done tractably when the conditional reward distribution and model prior are in the exponential family.

First, we assume that we can write the conditional reward likelihood as,

$$P(r \mid a, x, s; \theta) = \exp\left[\phi(r, a, x)^\top \eta(\theta_s) - \psi(a, x)^\top g(\theta_s)\right],$$

where $\phi(r, a, x)$ and $\psi(a, x)$ are sufficient statistics of the observed data, $\eta(\theta_s)$ are the natural parameters, and $g(\theta_s)$ satisfies $\psi(a, x)^\top g(\theta_s) = \log \sum_r \phi(r, a, x)^\top \eta(\theta_s)$, which is the log-partition function. Note that this assumption excludes rewards that are complex functions of the model parameters; however, it still handles all the non-contextual or linear bandit settings of prior work.

Then, we assume the prior over model parameters $\theta$ factors as $P_1(\theta) = \prod_{s \in \mathcal{S}} P_1(\theta_s)$. The prior over $\theta_s$ is assumed to be the conjugate prior of the likelihood and have the general form,

$$P_1(\theta_s) = h(\phi_{s,1}, m_{s,1}) \exp\left[\phi_{s,1}^\top \eta(\theta_s) - m_{s,1}^\top g(\theta_s)\right],$$

where $\phi_{s,1}, m_{s,1}$ are parameters controlling the prior, and $h(\phi_{s,1}, m_{s,1})$ is the normalizing factor.

Recall that $H_t = (X_\ell, A_\ell, R_\ell)_{\ell=1}^{t-1}$ is the history up to round $t$. For round $t$, we can write the joint posterior as,

$$P_t(s, \theta) = P_1(s) P_1(\theta) P(H_t \mid s; \theta)$$

$$\propto P_1(s) \left[\prod_{s' \neq s} P_1(\theta_{s'})\right] P_1(\theta_s) P(H_t \mid \theta_s) \propto P_1(s) \left[\prod_{s' \neq s} P_1(\theta_{s'})\right] P_t(\theta_s). \tag{7}$$

Given latent state $s$, we are only concerned with computing the posterior over the conditional model parameters $\theta_s$, denoted $P_t(\theta_s)$. This is given by,

$$P_t(\theta_s) \propto P_1(\theta_s) P(H_t \mid \theta_s) \tag{8}$$

$$\propto P_1(\theta_s) \prod_{\ell=1}^{t-1} \exp\left[\phi(R_\ell, A_\ell, X_\ell)^\top \eta(\theta_s) - \psi(A_\ell, X_\ell)^\top g(\theta_s)\right]$$

$$\propto h(\phi_{s,1}, m_{s,1}) \exp\left[\phi_{s,1}^\top \eta(\theta_s) - m_{s,1}^\top g(\theta_s) + \left(\sum_{\ell=1}^{t-1} \phi(R_\ell, A_\ell, X_\ell)\right)^\top \eta(\theta_s) - \left(\sum_{\ell=1}^{t-1} \psi(A_\ell, X_\ell)\right)^\top g(\theta_s)\right]$$

$$\propto h(\phi_{s,1}, m_{s,1}) \exp\left[\left(\phi_{s,1} + \sum_{\ell=1}^{t-1} \phi(R_\ell, A_\ell, X_\ell)\right)^\top \eta(\theta_s) - \left(m_{s,1} + \sum_{\ell=1}^{t-1} \psi(A_\ell, X_\ell)\right)^\top g(\theta_s)\right].$$

The general form for an exponential family likelihood is still retained. The prior-to-posterior conversion simply involves updating the prior parameters with sufficient statistics from the data. Specifically, updated parameters $\phi_{s,t} \leftarrow \phi_{s,1} + \sum_{\ell=1}^{t-1} \phi(R_\ell, A_\ell, X_\ell)$ and $m_{s,t} \leftarrow m_{s,1} + \sum_{\ell=1}^{t-1} \psi(A_\ell, X_\ell)$ form the conditional posterior $P_t(\theta_s) = h(\phi_{s,t}, m_{s,t}) \exp\left[\phi_{s,t}^\top \eta(\theta_s) - m_{s,t}^\top g(\theta_s)\right]$. Finally, the marginal posterior of $s$ is given by,

$$P_t(s) \propto P_1(s) P(H_t \mid s) \propto P_1(s) \int_{\theta_s} P_1(\theta_s) P(H_t \mid \theta_s) d\theta_s$$

$$\propto P_1(s) \int_{\theta_s} h(\phi_{s,1}, m_{s,1}) \exp\left[\phi_{s,t}^\top \eta(\theta_s) - m_{s,t} g(\theta_s)\right] d\theta_s \tag{9}$$

$$\propto P_1(s) \frac{h(\phi_{s,1}, m_{s,1})}{h(\phi_{s,t}, m_{s,t})}.$$

The ratio on the right can be easily computed by tracking the posterior parameters. In practice, the marginal posterior can also be updated incrementally as

$$
\begin{aligned}
P_t(s) &\propto P(s \mid H_{t-1})P(R_{t-1} \mid A_{t-1}, X_{t-1}, s) \\
&\propto P(s \mid H_{t-1}) \int_{\theta_s} P(\theta_s \mid H_{t-1})P(R_{t-1} \mid A_{t-1}, X_{t-1}, s; \theta)d\theta_s \\
&\propto P_{t-1}(s)\frac{h(\phi_{s,t-1}, m_{s,t-1})}{h(\phi_{s,t}, m_{s,t})} \,,
\end{aligned}
\tag{10}
$$

where the second term is effffectively the posterior predictive of the observations of round $t-1$.

Thus, for all states $s$, and parameters $\theta$, the posterior probabilities $P_t(s)$ and $P_t(\theta_s)$ have analytic, closed-form solutions. Thus, sampling from the joint posterior can be done tractably by sampling state $s$ from its marginal posterior, then conditional parameters $\theta_s$ from their posterior.

## B  Pseudocode of `mmTS` for Gaussians

Next, we provide specific variants of `mmTS` when both the model prior and conditional reward likelihood are Gaussian. This is a common assumption for Thompson sampling algorithms [3, 4, 2]. In this case, the joint posterior $P_t$ consists of Gaussians. We adopt the notation that $\mathcal{N}(r \mid \mu, \sigma^2) \propto \exp[-(r-\mu)^2/2\sigma^2]$ is the Gaussian likelihood of $r$ given mean $\mu$ and variance $\sigma^2$.

We detail algorithms for two cases: Algorithm 4 is for a multi-armed bandit with independent arms (no context), and Algorithm 5 is for a linear bandit problem. In the first case, we have that $\theta_s \in \mathbb{R}^K$ are the mean reward vectors where $\mu(a, s; \theta) = \theta_{s,a}$. In the other case, we assume that context is given by $x \in \mathbb{R}^{K \times d}$ where $x_a \in \mathbb{R}^d$ is the feature vector for arm $a$. Then, we have that $\theta_s \in \mathbb{R}^d$ are rank-$d$ vectors such that $\mu(a, x, s; \theta) = x_a^\top \theta_s$. Both algorithms are efficient to implement, and perform exact sampling from the joint posterior.

---

**Algorithm 4** Independent Gaussian mmTS (No Context)

---

1: **Input:**
2:     Prior over model parameters $P_1(\theta_s) = \mathcal{N}(\bar{\theta}_s, \sigma_0^2 I), \forall s \in \mathcal{S}$
3:     Prior over latent states $P_1(s)$

4: **for** $t \leftarrow 1, 2, \ldots$ **do**
5:     $\triangleright$ Step 1: sample latent state from marginal posterior.
6:     Sample $B_t \sim P_t$

7:     $\triangleright$ Step 2: sample model parameters from conditional posteriors.
8:     For each $s \in \mathcal{S}$, sample $\widehat{\theta}_s \sim \mathcal{N}(\tilde{\theta}_s, \mathrm{diag}(\tilde{\sigma}))$, where

$$\tilde{\sigma}_a \leftarrow \left(\sigma_0^{-2} + \sum_{\ell=1}^{t-1} \mathbb{1}\{A_\ell = a\}\, \sigma^{-2}\right)^{-1},$$

$$\tilde{\theta}_{s,a} \leftarrow \tilde{\sigma}_a \left(\sigma_0^{-2}\bar{\theta}_{s,a} + \sigma^{-2} \sum_{\ell=1}^{t-1} \mathbb{1}\{A_\ell = a\}\, R_\ell\right)$$

9:     Select $A_t \leftarrow \arg\max_{a \in A} \widehat{\theta}_{B_t,a}$

10:     $\triangleright$ Step 3: update marginal posterior.
11:     Observe $R_t$. Update

$$P_t(s) \propto P_{t-1}(s)\mathcal{N}(R_\ell \mid \tilde{\theta}_{s,A_\ell}, \tilde{\sigma}_{A_\ell} + \sigma^2)$$

---

---

**Algorithm 5** Linear Gaussian mmTS

---

1: **Input:**
2:     Prior over model parameters $P_1(\theta_s) = \mathcal{N}(\bar{\theta}_s, \Sigma_0), \forall s \in \mathcal{S}$
3:     Prior over latent states $P_1(s)$

4: **for** $t \leftarrow 1, 2, \ldots$ **do**
5:     $\triangleright$ Step 1: sample latent state from marginal posterior.
6:     Sample $B_t \sim P_t$

7:     $\triangleright$ Step 2: sample model parameters from conditional posteriors.
8:     Define

$$V_t \leftarrow I + \sum_{\ell=1}^{t-1} X_{\ell,A_\ell} X_{\ell,A_\ell}^\top, \qquad F_t \leftarrow \sum_{\ell=1}^{t-1} X_{\ell,A_\ell} R_\ell$$

9:     For each $s \in \mathcal{S}$, compute $\widehat{\beta} \leftarrow V_t^{-1} F_t$, and $\widehat{\Sigma} \leftarrow \sigma^2 V_t^{-1}$
10:     For each $s \in \mathcal{S}$, sample $\widehat{\theta}_s \sim \mathcal{N}(\tilde{\theta}_s, \tilde{\Sigma})$, where

$$\tilde{\Sigma} \leftarrow \left(\Sigma_0^{-1} + (t-1)\widehat{\Sigma}^{-1}\right)^{-1}, \qquad \tilde{\theta}_s \leftarrow \tilde{\Sigma}\left(\Sigma_0^{-1}\bar{\theta}_s + (t-1)\widehat{\Sigma}^{-1}\widehat{\beta}\right)$$

11:     Select $A_t \leftarrow \arg\max_{a \in A} X_{\ell,a}^\top \widehat{\theta}_{B_t}$

12:     $\triangleright$ Step 3: update marginal posterior.
13:     Observe $R_t$. Update

$$P_t(s) \propto P_{t-1}(s)\mathcal{N}(R_\ell \mid X_{\ell,A_\ell}^\top \tilde{\theta}_s, X_{\ell,A_\ell}^\top \tilde{\Sigma} X_{\ell,A_\ell} + \sigma^2)$$

---

## C Proofs

Our proofs rely on the following concentration inequality, which is a straightforward extension of the Azuma-Hoeffding inequality to sub-Gaussian random variables.

**Lemma 1.** *Let $(Y_t)_{t \in [n]}$ be a martingale difference sequence with respect to filtration $(\mathcal{F}_t)_{t \in [n]}$, that is $\mathbb{E}\left[Y_t \mid \mathcal{F}_{t-1}\right] = 0$ for any $t \in [n]$. Let $Y_t \mid \mathcal{F}_{t-1}$ be $\sigma^2$-sub-Gaussian for any $t \in [n]$. Then for any $\varepsilon > 0$,*

$$\mathbb{P}\left(\left|\sum_{t=1}^{n} Y_t\right| \geq \varepsilon\right) \leq 2\exp\left[-\frac{\varepsilon^2}{2n\sigma^2}\right].$$

*Proof.* For any $\lambda > 0$, which we tune later, we have

$$\mathbb{P}\left(\sum_{t=1}^{n} Y_t \geq \varepsilon\right) = \mathbb{P}\left(\prod_{t=1}^{n} e^{\lambda Y_t} \geq e^{\lambda \varepsilon}\right) \leq e^{-\lambda \varepsilon} \mathbb{E}\left[\prod_{t=1}^{n} e^{\lambda Y_t}\right].$$

The inequality is by Markov's inequality. From the conditional independence of $Y_t$ given $\mathcal{F}_{t-1}$, the right term becomes

$$\mathbb{E}\left[\prod_{t=1}^{n} e^{\lambda Y_t}\right] = \mathbb{E}\left[\mathbb{E}\left[e^{\lambda Y_n} \mid \mathcal{F}_{n-1}\right]\prod_{t=1}^{n-1} e^{\lambda Y_t}\right] \leq e^{\frac{\lambda^2 \sigma^2}{2}}\mathbb{E}\left[\prod_{t=1}^{n-1} e^{\lambda Y_t}\right] \leq e^{\frac{n\lambda^2 \sigma^2}{2}}.$$

We use that $Y_n \mid \mathcal{F}_{n-1}$ is $\sigma^2$-sub-Gaussian in the first inequality, and recursively repeat for all rounds in the second. So we have

$$\mathbb{P}\left(\sum_{t=1}^{n} Y_t \geq \varepsilon\right) \leq \min_{\lambda > 0} e^{-\lambda \varepsilon + \frac{n\lambda^2 \sigma^2}{2}}.$$

The minimum is achieved at $\lambda = \varepsilon/(n\sigma^2)$. Therefore

$$\mathbb{P}\left(\sum_{t=1}^{n} Y_t \geq \varepsilon\right) \leq \exp\left[-\frac{\varepsilon^2}{2n\sigma^2}\right].$$

Now we apply the same proof to $\mathbb{P}\left(-\sum_{t=1}^{n} Y_t \geq \varepsilon\right)$, which yields a multiplicative factor of 2 in the upper bound. This concludes the proof. $\square$

### C.1 Proof of Theorem 1

Recall that $s_* \in \mathcal{S}, \theta_* \in \Theta$ are the true latent state and model. Let $\mu(a, x) = \mu(a, x, s_*; \theta_*)$ be the true mean rewards given observed context and action. Let

$$E_t = \left\{\forall s \in \mathcal{S} : \left|\sum_{\ell=1}^{t-1} \mathbb{1}\{B_\ell = s\}\left(\mu(A_\ell, X_\ell) - R_\ell\right)\right| \leq \sigma\sqrt{6N_t(s)\log n}\right\} \tag{11}$$

be the event that the total realized reward under each played latent state is close to its expectation. Let $E = \cap_{t=1}^{n} E_t$ be the event that this holds for all rounds, and $\bar{E}$ be its complement. We can rewrite the expected $n$-round regret by

$$\mathcal{R}(n) = \mathbb{E}\left[\mathbb{1}\{\bar{E}\}\mathcal{R}(n)\right] + \mathbb{E}\left[\mathbb{1}\{E\}\mathcal{R}(n)\right]$$

$$\leq \mathbb{E}\left[\mathbb{1}\{\bar{E}\}\sum_{t=1}^{n} \mu(A_{t,*}, X_t) - \mu(A_t, X_t)\right]$$

$$+ \mathbb{E}\left[\mathbb{1}\{E\}\sum_{t=1}^{n}\left(\mu(A_{t,*}, X_t) - U_t(A_{t,*})\right)\right] + \mathbb{E}\left[\mathbb{1}\{E\}\sum_{t=1}^{n}\left(U_t(A_t) - \mu(A_t, X_t)\right)\right], \tag{12}$$

where we use the regret decomposition in Eq. (5) in the inequality.

Our first lemma is that the probability of $\bar{E}$ occurring is low. Without context, the lemma would follow immediately from Hoeffding's inequality. Since we have context generated by some random process, we instead turn to martingales.

**Lemma 2.** *Let $E_t$ be defined as in Eq. (11) for all rounds $t$, $E = \cap_{t=1}^n E_t$, and $\bar{E}$ be its complement. Then $\mathbb{P}\left(\bar{E}\right) \leq 2|\mathcal{S}|n^{-1}$.*

*Proof.* We see that the choice of action given observed context depends on past rounds. This is because the upper confidence bounds depend on which latent states are eliminated, which depend on the history of observed contexts.

For each latent state $s$ and round $t$, let $\mathcal{T}_{t,s}$ be the rounds where state $s$ was chosen until round $t$. For round $\ell \in \mathcal{T}_{t,s}$, let $Y_\ell(s) = \mu(A_\ell, X_\ell, s_\ell; \theta_*) - R_\ell$. Observe that $Y_\ell(s) \mid X_\ell, H_\ell$ is $\sigma^2$-sub-Gaussian. This implies that $(Y_\ell(s))_{\ell \in \mathcal{T}_{t,s}}$ is a martingale difference sequence with respect to context and history $(X_\ell, H_\ell)_{\ell \in \mathcal{T}_{t,s}}$, or $\mathbb{E}\left[Y_\ell(s) \mid X_\ell, H_\ell\right] = 0$ for all rounds $\ell \in \mathcal{T}_{t,s}$.

For any round $t$, and state $s \in \mathcal{S}$, we have that $\mathcal{T}_{t,s}$ is a random quantity. First, we fix $|\mathcal{T}_{t,s}| = N_t(s) = u$ where $u < t$ and yield the following due to Lemma 1,

$$\mathbb{P}\left(\left|\sum_{\ell \in \mathcal{T}_{t,s}} Y_\ell(s)\right| \geq \sigma\sqrt{6u \log n}\right) \leq 2\exp\left[-3\log n\right] = 2n^{-3}.$$

So, by the union bound, we have

$$\mathbb{P}\left(\bar{E}\right) \leq \sum_{t=1}^n \sum_{s \in \mathcal{S}} \sum_{u=1}^{t-1} \mathbb{P}\left(\left|\sum_{\ell \in \mathcal{T}_{t,s}} Y_\ell(s)\right| \geq \sigma\sqrt{6u \log n}\right) \leq 2|\mathcal{S}|n^{-1}.$$

$\square$

The first term in Eq. (12) is small because the probability of $\bar{E}$ is small. Using Lemma 2, and that total regret is bounded by $n$, we have, $\mathbb{E}\left[\mathbb{1}\{\bar{E}\}\mathcal{R}(n)\right] \leq n\mathbb{P}\left(\bar{E}\right) \leq 2|\mathcal{S}|$.

For round $t$, the event $\mu(A_{t,*}, X_t) \geq U_t(A_{t,*})$ occurs only if $s_* \notin C_t$ also occurs. By the design of $C_t$ in mUCB, this happens only if $G_t(s_*) > \sigma\sqrt{6N_t(s)\log n}$. Event $E_t$ says that the opposite is true for all states, including true state $s_*$. So, the second term in Eq. (12) is at most 0.

Now, consider the last term in Eq. (12). Let $T_s = \{t \leq n : B_t = s\}$ denote the set of rounds where latent state $s$ is selected. We have,

$$\mathbb{E}\left[\mathbb{1}\{E\}\sum_{t=1}^n \left(U_t(A_t) - \mu(A_t, X_t)\right)\right] = \mathbb{E}\left[\mathbb{1}\{E\}\sum_{s \in S}\sum_{t \in T_s}\left(\mu(A_t, X_t, s) - \mu(A_t, X_t)\right)\right]$$

$$= \mathbb{E}\left[\mathbb{1}\{E\}\sum_{s \in S}\sum_{t \in T_s}\left(\mu(A_t, X_t, s) - R_t + R_t - \mu(A_t, X_t)\right)\right]$$

$$\leq \mathbb{E}\left[\mathbb{1}\{E\}\sum_{s \in S}\left(G_n(s) + \sum_{t \in T_s}\left(R_t - \mu(A_t, X_t)\right)\right)\right]$$

$$\leq \sum_{s \in S}\left(1 + 2\sigma\sqrt{6N_n(s)\log n}\right).$$

For the first inequality, we use that the last round $t'$ where state $s$ is selected, we have an upper-bound on the prior gap $G_{t'}(s) \leq \sigma\sqrt{6N_{t'}(s)\log n}$. Accounting for the last round yields an upper-bound of $\sigma\sqrt{6N_n(s)\log n} + 1$. For the last inequality, we use $E$ occurring to bound $\sum_{t \in T_s}\left(R_t - \mu(A_t, X_t)\right) \leq \sigma\sqrt{6N_n(s)\log n}$.

This yields the desired bound on total regret,

$$\mathcal{R}(n) \leq 3|\mathcal{S}| + 2\sigma\sqrt{6\log n}\left(\sum_{s \in S}\sqrt{N_n(s)}\right)$$

$$\leq 3|\mathcal{S}| + 2\sigma\sqrt{6|\mathcal{S}|\log n \sum_{s \in S}N_n(s)} = 3|\mathcal{S}| + 2\sigma\sqrt{6|\mathcal{S}|n\log n},$$

where the last inequality comes from the Cauchy–Schwarz inequality.

## C.2 Proof of Corollary 1

The true latent state $S_* \in \mathcal{S}$ is random under Bayes regret. In this case, we still assume that we are given the true model $\theta_*$, so only $S_* \sim P_1$ for known $P_1$. We also have that the optimal action $A_{t,*} = \arg\max_{a \in \mathcal{A}} \mu(a, X_t, S_*; \theta_*)$ is random not only due to context, but also $S_*$.

We define $U_t(a) = \arg\max_{s \in C_t} \mu(a, X_t, S_*; \theta_*)$ as in mUCB. Note the additional randomness due to $S_*$. We can rewrite the Bayes regret as $\mathcal{BR}(n) = \mathbb{E}\left[\mathcal{R}(n; S_*, \theta_*)\right]$, where the outer expectation is over $S_* \sim P_1$. The expression inside the expectation can be decomposed as

$$
\begin{aligned}
\mathcal{R}(n, S_*, \theta_*) = & \mathbb{E}\left[\mathbb{1}\{\bar{E}\} \sum_{t=1}^{n} \mu(A_{t,*}, X_t, S_*) - \mu(A_t, X_t, S_*)\right] \\
& + \mathbb{E}\left[\mathbb{1}\{E\} \sum_{t=1}^{n} (\mu(A_{t,*}, X_t, S_*) - U_t(A_{t,*}))\right] + \mathbb{E}\left[\mathbb{1}\{E\} \sum_{t=1}^{n} (U_t(A_t) - \mu(A_t, X_t, S_*))\right],
\end{aligned}
$$

where $E, \bar{E}$ are defined as in Appendix C.1, and we use the decomposition in Eq. (6).

Each above expression can be bounded exactly as in Theorem 1. The reason is that the original upper bounds hold for any $S_*$, and therefore also in expectation over $S_* \sim P_1$. This yields the desired Bayes regret bound.

## C.3 Proof of Theorem 2

The only difference in the analysis is that we need to incorporate the additional error due to model misspecification.

Let $\mathcal{E} = \{\forall a \in \mathcal{A}, x \in \mathcal{X}, s \in \mathcal{S} : |\widehat{\mu}(a, x, s) - \mu(a, x, s)| \leq \varepsilon\}$ be the event that model $\widehat{\theta}$ has bounded misspecification and $\bar{\mathcal{E}}$ be its complement. Also let $E, \bar{E}$ be defined as in Appendix C.1.

If $\mathcal{E}$ does not hold, then the best possible upper-bound on regret is $n$; fortunately, we assume in the theorem that the probability of that occurring is bounded by $\delta$. So we can bound the expected $n$-round regret as

$$
\begin{aligned}
\mathcal{R}(n) = & \mathbb{E}\left[\mathbb{1}\{\bar{\mathcal{E}}\} \mathcal{R}(n)\right] + \mathbb{E}\left[\mathbb{1}\{\bar{E}, \mathcal{E}\} \mathcal{R}(n)\right] + \mathbb{E}\left[\mathbb{1}\{E, \mathcal{E}\} \mathcal{R}(n)\right] \\
\leq & \delta n + \mathbb{E}\left[\mathbb{1}\{\bar{E}, \mathcal{E}\} \sum_{t=1}^{n} \mu(A_{t,*}, X_t) - \mu(A_t, X_t)\right] \\
& + \mathbb{E}\left[\mathbb{1}\{E, \mathcal{E}\} \sum_{t=1}^{n} (\mu(A_{t,*}, X_t) - U_t(A_{t,*}))\right] + \mathbb{E}\left[\mathbb{1}\{E, \mathcal{E}\} \sum_{t=1}^{n} (U_t(A_t) - \mu(A_t, X_t))\right],
\end{aligned}
\tag{13}
$$

where we use the regret decomposition in Eq. (5).

The second term in Eq. (13) is small because the probability of $\bar{E}$ is small. Using Lemma 2, and that total regret is bounded by $n$, we have, $\mathbb{E}\left[\mathbb{1}\{\bar{E}, \mathcal{E}\} \mathcal{R}(n)\right] \leq n \mathbb{P}\left(\bar{E}\right) \leq 2|\mathcal{S}|$.

If $\mathcal{E}$ occurs, the event $\mu(A_{t,*}, X_t) - U_t(A_{t,*}) > \varepsilon$ for any round $t$ occurs only if $s_* \notin C_t$ also occurs. By the design of $C_t$ in mmUCB, this happens if $G_t(s_*) \geq \sigma\sqrt{6N_t(s)\log n}$. Since

$$
G_t(s_*) = \sum_{\ell=1}^{t-1} \mathbb{1}\{B_\ell = s_*\} (\widehat{\mu}(A_\ell, X_\ell) - \varepsilon - R_\ell) \leq \sum_{\ell=1}^{t-1} \mathbb{1}\{B_\ell = s_*\} (\mu(A_\ell, X_\ell) - R_\ell),
$$

we see that event $E_t$ says that the opposite is true for all states, including true state $s_*$. Hence, the third term in Eq. (13) is bounded by $\varepsilon n$.

Now, consider the last term in Eq. (13). Let $T_s = \{t \le n : B_t = s\}$ denote the set of rounds where latent state $s$ is selected. We have,

$$\mathbb{E}\left[\mathbb{1}\{E, \mathcal{E}\} \sum_{t=1}^{n} (U_t(A_t) - \mu(A_t, X_t))\right]$$

$$= \mathbb{E}\left[\mathbb{1}\{E, \mathcal{E}\} \sum_{s \in S} \left(\sum_{t \in T_s} \widehat{\mu}(A_t, X_t, s) - \mu(A_t, X_t)\right)\right]$$

$$= \varepsilon n + \mathbb{E}\left[\mathbb{1}\{E, \mathcal{E}\} \sum_{s \in S} \sum_{t \in T_s} (\widehat{\mu}(A_t, X_t, s) - \varepsilon - R_t + R_t - \mu(A_t, X_t))\right]$$

$$\le \varepsilon n + \mathbb{E}\left[\mathbb{1}\{E, \mathcal{E}\} \sum_{s \in S} \left(G_n(s) + \sum_{t \in T_s} (R_t - \mu(A_t, X_t))\right)\right]$$

$$\le \varepsilon n + \sum_{s \in S} \left(1 + 2\sigma\sqrt{6 N_n(s) \log n}\right).$$

For the first inequality, we use that the last round $t'$ where state $s$ is selected, we have an upper-bound on the prior gap $G_{t'}(s) \le \sigma\sqrt{6 N_{t'}(s) \log n}$. Accounting for the last round yields and upper-bound of $\sigma\sqrt{6 N_n(s) \log n} + 1$. For the last inequality, we use $E$ occurring to bound $\sum_{t \in T_s} (R_t - \mu(A_t, X_t)) \le \sigma\sqrt{6 N_n(s) \log n}$.

This yields the desired bound on total regret,

$$\mathcal{R}(n) \le \delta n + 3|\mathcal{S}| + 2\varepsilon n + 2\sigma\sqrt{6 \log n}\left(\sum_{s \in S} \sqrt{N_n(s)}\right)$$

$$\le \delta n + 3|\mathcal{S}| + 2\varepsilon n + 2\sigma\sqrt{6|\mathcal{S}| \log n \sum_{s \in S} N_n(s)} = \delta n + 3|\mathcal{S}| + 2\varepsilon n + 2\sigma\sqrt{6|\mathcal{S}|n \log n},$$

where the last inequality comes from the Cauchy–Schwarz inequality.

### C.4 Proof of Corollary 2

Both latent state $S_* \in \mathcal{S}$ and model $\theta_* \in \Theta$ are random, and drawn as $S_*, \theta_* \sim P_1$, where the prior $P_1$ is known. In this case, the true model $\theta_*$ is not known to us.

Using marginalized means $\bar{\mu}(a, x, s)$, and $\varepsilon, \delta > 0$ as defined in the statement of the corollary, we write,

$$G_t(s) = \sum_{\ell=1}^{t-1} \mathbb{1}\{B_\ell = s\} (\bar{\mu}(A_\ell, X_\ell, s) - \varepsilon - R_\ell),$$

and $U_t(a) = \arg\max_{s \in C_t} \bar{\mu}(a, X_t, s)$. This is in contrast to $G_t(s)$ and $U_t(a)$ in mmUCB, which use $\widehat{\mu}(a, x, s)$ from a single model. Conceptually though, both $\widehat{\mu}(a, x, s)$ and $\bar{\mu}(a, x, s)$ are just $\varepsilon$-close point estimates of $\mu(a, x, s)$ due to the assumptions made about the true model $\theta_*$ in the theorem and corollary, respectively.

We can rewrite the Bayes regret as $\mathcal{BR}(n) = \mathbb{E}\left[\mathcal{R}(n; S_*, \theta_*)\right]$, where the outer expectation is over $S_*, \theta_* \sim P_1$. The expression inside the expectation can be written as,

$$\mathcal{R}(n; S_*, \theta_*) \le \delta n + \mathbb{E}\left[\mathbb{1}\{\bar{E}, \mathcal{E}\} \sum_{t=1}^{n} \mu(A_{t,*}, X_t, S_*; \theta_*) - \mu(A_t, X_t, S_*; \theta_*)\right]$$

$$+ \mathbb{E}\left[\mathbb{1}\{E, \mathcal{E}\} \sum_{t=1}^{n} (\mu(A_{t,*}, X_t, S_*; \theta_*) - U_t(A_{t,*}))\right] + \mathbb{E}\left[\mathbb{1}\{E, \mathcal{E}\} \sum_{t=1}^{n} (U_t(A_t) - \mu(A_t, X_t, S_*; \theta_*))\right],$$

where $\mathcal{E}, E, \bar{E}$ are defined as in Appendix C.3, and we use the decomposition in Eq. (6).

The expressions can be bounded exactly as in Theorem 2. The upper bound is worst-case and holds for any $S_*, \theta_*$, and thus also holds after taking an expectation over the prior $S_*, \theta_* \sim P_1$. This bounds the Bayes regret, as desired.