[Reviews · NeurIPS 2020]

Review 1

Summary and Contributions: The paper considers the latent bandits with context information, where the parameter vector is available from offline evaluations (either exactly or approximately). The paper proposes both UCB and Thompson Sampling-based algorithms for the problem and demonstrates their efficacy via synthetic/real-world experiments

Strengths: The main strength I believe lies in the practicality of the setting. Often a parameter estimate is available from offline estimates and the ideas proposed (m2 variants) use these carefully to develop efficient algorithms.

Weaknesses: While the authors perform empirical evaluation on synthetic and real world data, they seem to restrict themselves to "fast personalization" horizons. The algorithm per se is not designed keeping short horizons in mind. It appears from the experimental results that the general trend indicates that LinTS would perform better than m2TS if the horizon was extended a bit more. It would be nice if the authors can discuss this a bit more.

Correctness: The empircal methodology is reasonable. The authors give a slightly corrected version of their proofs in the supplementary material (where the constants change). Other than that, the overall proof techniques seem to be standard.

Clarity: The paper is well written and is clear.

Relation to Prior Work: Looks reasonable.

Reproducibility: Yes

Additional Feedback:


Review 2

Summary and Contributions: The paper considers a bandit setting with a latent space: each user is associated to a state which controls her preferences. The setting assumes an approximate version of the latent space is given by some offline algorithm. Therefore, the bandit algorithm is aiming at identifying the state of the user which corresponds the most to her preferences. 4 algorithms are proposed: * 2 based on upper confidence bounds (UCB), 2 based on Thompson Sampling (TS) approach; * 2 assuming knowledge of the exact latent space, 2 assuming an approximation thereof. A proof of an upper-bound on the worst case regret is given for the UCB-based algorithms and a proof of an upper-bound on the bayesian regret is given for TS-based algorithms. Some experiments compare the result of the proposed algorithm vs. state of the art algorithms making no use of the latent model, or using it in a conservative way. --- Post Rebuttal --- Thank you for your precise remarks. I'm eager to look at a more refined analysis with respect to epsilon; either in the context where mmTS posterior reduces the impact of epsilon, or in the entire bandit setting which would include the offline learning. Note that Zhou and Brunskill (2016) handle the entire setting by adding a uniform exploration for the \tau_u first steps of each sequence of recommendations, which was sufficient to control the evolution of epsilon along successive sequences (under some assumption on the offline learning problem).

Strengths: The work presents and analyzes in a unified way a set of approaches (UCB-like and TS-like), of settings (independent arms and contextual bandit), and of assumptions (exact and approximate latent space).

Weaknesses: First, the approach puts appart the offline learning of the latent factors. While this learning build upon much more data, it also suffers an exploration-exploitation tradeoff. Secondly, the unified analyses cannot cover instance-dependent bounds of the regret which inform about the settings which are harder to handle.

Correctness: yes

Clarity: yes

Relation to Prior Work: yes

Reproducibility: Yes

Additional Feedback:


Review 3

Summary and Contributions: The paper adds to the algorithmic and regret analysis of the contextual latent bandits model of Zhou and Brunskill (itself based on work by Maillard and Mannor), which is designed to allow quick adaptation of bandit models to user characteristics. The paper adds a conservative correction to deal with model error and explores the use of Thompson sampling as an alternative to UCB algorithms.

Strengths: A cleanly written and technically competent paper with analysis and experimental results suggesting that the approach could be useful in practice. Does a good job of motivating the model. The "pedagogical" experiments nicely illustrate the benefits of the algorithmic contributions.

Weaknesses: The algorithms are essentially standard and the results are somewhat incremental. The algorithm changes and analysis that allow for epsilon-inaccurate conditional models are fairly minor. The regret analysis for Thompson sampling follows the lines laid out by Van Roy. From a technical point of view, the possible weak point is in the need to supply an epsilon to the m^2UCB algorithm. Often, the available epsilon-bounds are wildly pessimistic and would render the algorithm useless, so you'd have to just tweak it experimentally to get something to work. Similar issue with a misspecified hyperprior P_1 for the m^2TS algorithm. The paper makes no attempt to analyze the problem in computational terms: how hard is it? Could one compute an optimal policy? If that could be done even for small cases, how does your algorithm compare to optimality in practice? (The graphs show "optimal", but that's post hoc clairvoyant.) The algorithmic ideas here are all inherited from bandit algorithms, but this is not really an arm-space bandit problem from the computational viewpoint because the reward models are known.

Correctness: The theorems appear to be correct. The paper could do better than gloss over the epsilon/hyperprior issue mentioned above.

Clarity: The paper is very clear and maintains the right level of detail in the main body. Related to the point that this is also a value-of-information problem, might be useful to show the setup as a Bayes net or plate diagram Minor comments: l.30 adapt it results -> adapt its results l.49 "its effectiveness" - referent is "algorithms" or "approaches", so plural? l.77 "offline interaction data" is an oxymoronic phrase; I assume you mean repositories of interaction data? l.115 "An advantage of TS over UCB is that it obviates the need to design UCBs" - hmmmm. l.203 i.e. tensor decomposition -> e.g., tensor decomposition l.204 as size of offline dataset -> as the size of the offline dataset l.218 no offline -> no offline model l.218 "using offline reward model as experts" ?? rewrite l.226 "non-" ?? They have 10 arms. Do you mean non-contextual? l.227 uniformly at random -> uniformly at random: l.321 yadkori -> Yadkori [check the pdf!] l.339 thompson -> Thompson Missing sources in [16] and [30]

Relation to Prior Work: Does a good job within the narrow area of latent/contextual bandits. This has become a fairly dense area these days so I could be missing something. l.21 "explicit exploration for (latent) state identification (i.e., reducing uncertainty regarding the true state) is less common in practice." - Is this really true? It seems close to a standard value-of-information problem. The "bandit" aspect seems nearly orthogonal. Possibly some of the many results by Krause and collaborators on optimal and near-optimal information-gathering would apply here.

Reproducibility: Yes

Additional Feedback: Re broader impacts: content selection algorithms that explicitly aim to identify user "type" could exacerbate stereotyping and lead to enhanced psychological manipulation if combined with RL-like methods that learn to apply content *sequences* to change user mental state in profit-maximizing directions.


Review 4

Summary and Contributions: This paper provides new algorithms for contextual multi-arm bandits where reward depends not only on a context x and an action (or arm) a but also on a latent variable s, which is assumed to be unknown. Both UCB and Thompson sampling algorithms are proposed and regret bounds are derived.

Strengths: - The problem of "Latent bandits" is useful as often the some useful information (e.g. intent behind action) in off-line data is not known. - The paper is well-written and provides a theoretical analysis of the algorithm.

Weaknesses: Given the previous works such as Latent Bandit [23] and the connections drawn between UCB and TS in Russo and Van Roy [26], the results of this paper, especially incorporating model uncertainty in the latent bandits seems straightforward. Therefore, novelty is weak. Proof techniques are also heavily borrowed from the previous papers.

Correctness: As far as I can tell the claims made and methods used in this paper are correct, though I have not checked all details for the proofs in the supplementary.

Clarity: The paper is easy to read and usually clear except the problem setting for the MovieLens experiments. See my comments below.

Relation to Prior Work: The paper discusses some closely related works but it could be better if it could also cover the work on latent bandits with confounding information and its relation with them.

Reproducibility: No

Additional Feedback: Comment on the proposed algorithm: Overall, the algorithm seems reasonable and is simple. I have the following questions: - The regret bounds for the misspecified model case (Theorem 2 and Corollary 2) are linear in n, does this mean the proposed algorithm is not efficient in this case? For a fixed size off-line dataset, this could be a problem. - Does the algorithm assume the latent state to be fixed throughout the optimisation or can it be changed over time? - What about the number of latent states, it seems the number needs to be known in advance? If not so, it is not clear how will the proposed algorithm tackle that case? Comments on the experiments: The proposed methods are compared with some methods with no offline model, and Exp4. While Exp4 performance seems a bit surprising, I am wondering how will the proposed methods compare with a simple baseline where latent contexts are identified via an online clustering and then we simply use the traditional contextual MAB algorithms on top of clustering? Further, to measure its effectiveness, the proposed method needs to be compared with the latent bandit paper in [23]. For the MovieLens experiments, I could not understand the problem setting properly, it is not clear to me in this setting what is the context, what is action and what is a latent state? This needs to be clearly described. Minor points: -The definition of \mu function is not consistent, e.g. we have \mu(a,x,s) or \mu(a,x,s,\theta), is it on purpose? -Line 164: Should s* be s? -Line 30: "it" should be "its" *** Post rebuttal *** After reading the rebuttal, I have updated my score.

[Author Response · NeurIPS 2020]

We would like to thank the reviewers for their insightful reviews. We are glad that the feedback was generally positive, as the reviewers were happy with the motivation and practicality of our work.

The primary weakness that several reviewers brought up was that the methods and analysis were straightforward. We believe that our novelty is in proposing Thompson sampling latent bandit algorithms using offline-learned graphical models (PGMs) as side information, with full regret analysis. Reviewers are correct in noting that our analysis relies on insights made earlier by Russo and Van Roy. However, we are the first to apply it to bandit problems with latent variables, which is very common in many applications e.g., personalized recommender systems. Our near-optimal analysis is very general and we are working on applying it to problems with more complicated and problem-dependent PGMs (e.g., PGMs with latent state transition dynamics or with factored, continuous, latent state structures).

**Reviewer #1** *"algorithm ... is not designed keeping short horizons in mind"*: Our algorithms quickly personalize by assuming users can be clustered into a finite set of latent states. When the set is small, identifying which state best describes the user's preferences can happen much quicker than learning the user's preferences from scratch.

**Reviewer #2** *"suffers an exploration-exploitation tradeoff"*: You are correct in noting that our algorithm depends on exploration of the offline data to learn good models. If the models are good though, our method achieves much greater sample-efficiency than baselines that ignore the offline data.

*"unified analyses cannot cover instance-dependent bounds"*: We derive Bayes regret bounds, which contain an expectation over possible instances. We agree that instance-dependent regret bounds are more informative, but more difficult to derive and interesting future work. Our view our work as a first step to achieving this.

**Reviewer #3** Thank you for your detailed corrections! We will update the paper with your clarifications.

*"the available epsilon-bounds are wildly pessimistic"*: You are correct in noting that our regret bounds require that the offline-learned model has low prediction error $\varepsilon$. Choosing $\varepsilon$ using model-learning guarantees will likely lead to an overly conservative mmUCB algorithm. However, we view mmUCB not as a practical algorithm, but one that can be used to analyze mmTS, which is practical and general. The mmTS algorithm additionally computes posterior model parameters using online interactions. This means the $\varepsilon$ term should not affect its long-term performance, and the regret bound we derived is likely too conservative. Updating our regret bounds to reflect this is a future line of work.

*"no attempt to analyze the problem in computational terms"*: Given a latent variable model, the optimal policy would plan out the entire sequence of actions using the model, akin to solving a special case of a POMDP. It is unclear how to do so feasibly in our setting, even on short horizons. Gittins index will compute an optimal strategy for Bayesian bandits, but we are also unsure how to generalize the method to complex latent models. We instead compare our algorithms to a "post hoc clairvoyant" algorithm that performs at least as well

*Relation to value-of-information problem*: A key distinction is that prior value-of-information work seems more concerned with discovering the optimal solution, similar to best-arm-identification, whereas our setting deals with regret minimization. We will cite such work by Krause et. al and discuss its differences in our work.

**Reviewer #4** *"regret bounds for the misspecified model case ... are linear in n"*: It is true that the performance of our algorithms depend on the quality of the offline-learned model. Past work in offline model-learning e.g. spectral methods, can give guarantees that are $\varepsilon = O(1/\sqrt{n_{\text{offline}}})$ where $n_{\text{offline}}$ is the size of the offline dataset. On short horizons $n \leq n_{\text{offline}}$, the contribution of $\varepsilon$ to overall regret is small. It is also important to note that our proposed mmTS algorithm computes posterior model parameters given online interactions. This means the $\varepsilon$ term due to the prior is too conservative. Having a regret bound that reflects this property remains as future work.

*Questions about latent states*: We consider the latent bandits setting in Maillard and Mannor where the underlying latent state is fixed, and comes from a finite set of known size. In practice, the number of latent states could be be tuned during offline learning via cross validation. Our algorithms can also work with non-parametric latent models, though parametric ones are more well-studied in literature and have recovery guarantees.

*Questions about MovieLens experiment*: For each episode/user, actions are movies to recommend, and context is the concatenation of feature vectors for each movie; the context was learned from the training set and known beforehand. The mean reward for recommending movie $i$ to user $j$ is the dot product between movie $i$ and user $j$'s feature vectors, both of which derived from the test set and not known to the learning method. Our hypothesis was that using a Gaussian mixture-model (GMM) over user features fitted on the train set would improve performance by allowing for quickly associating each user's hidden preferences with a cluster in the GMM. We will clarify the ambiguities in our work.

*"Exp4 performance seems a bit surprising"*: Exp4 can solve the latent bandits setting if we interpret the conditional model for each latent state as an expert; however, Exp4 is designed for adversarial rather than stochastic settings, which is why our algorithms greatly outperformed Exp4 using the same GMM model.

[Meta-Review · NeurIPS 2020]

All reviewers are inclined towards acceptance, primarily because of the clear connection to practical settings where some a priori model information is available, development of a Thompson sampling algorithm and its analysis along with natural optimism-based strategies, coverage of misspecification of the latent space, and a reasonably comprehensive experimental evaluation of latent bandit algorithms. Hence I recommend acceptance. However, from some of my earlier readings of prior work on latent bandits, I am not convinced about the validity of the remark "The closest work to ours is that of 278 Maillard and Mannor [23], which proposes and analyzes non-contextual UCB algorithms under the 279 assumption that the mean rewards for each latent state are known"; from what I have seen, M&M actually do not assume the mean rewards are known (e.g., A-UCB strategy). Moreover, the "B" sets play the exact same role as a context, and the Cs are the latent/unknown classes (called s in this paper). So I urge the author(s) to take a closer look at prior work and fix the comparison to related work in a careful manner.